# Refinement of an Established Procedure and Its Application for Identification of Hypoxia in Prostate Cancer Xenografts

**DOI:** 10.3390/cancers13112602

**Published:** 2021-05-26

**Authors:** Pernille B. Elming, Thomas R. Wittenborn, Morten Busk, Brita S. Sørensen, Mathilde Borg Houlberg Thomsen, Trine Strandgaard, Lars Dyrskjøt, Steffen Nielsen, Michael R. Horsman

**Affiliations:** 1Experimental Clinical Oncology-Department of Oncology, Aarhus University Hospital, 8200 Aarhus, Denmark; Wittenborn@biomed.au.dk (T.R.W.); bsin@clin.au.dk (B.S.S.); steffen.nielsen89@gmail.com (S.N.); mike@oncology.au.dk (M.R.H.); 2Danish Center for Particle Therapy, Aarhus University Hospital, 8200 Aarhus, Denmark; morten@oncology.au.dk; 3Department of Molecular Medicine, Aarhus University Hospital, 8200 Aarhus, Denmark; m_thomsen85@hotmail.com (M.B.H.T.); ts@clin.au.dk (T.S.); lars@clin.au.dk (L.D.); 4Department of Clinical Medicine, Aarhus University, 8200 Aarhus, Denmark

**Keywords:** hypoxia, pre-clinical models, prostate cancer, PC3, DU-145, hypoxia gene signature

## Abstract

**Simple Summary:**

Regions of low oxygen status (hypoxia) are a characteristic feature of solid tumors. This hypoxia is a major cause of resistance to treatment, especially radiation therapy, and it can increase the likelihood of metastatic spread. Being able to accurately identify tumor hypoxia will allow us to predict a patient’s response to therapy and find alternative approaches to improve outcome. We have refined a dissection method that involves autoradiography and laser-guided microdissection of hypoxic areas in tumors. Using this approach, we were able to test the feasibility of applying a 15-gene signature that was previously developed for head and neck cancer patients to identify hypoxia in pre-clinical models of prostate cancer. Our results demonstrated the potential of this method to identify hypoxia in this tumor type and suggest its applicability for use in patients with prostate cancer.

**Abstract:**

Background: This pre-clinical study was designed to refine a dissection method for validating the use of a 15-gene hypoxia classifier, which was previously established for head and neck squamous cell carcinoma (HNSCC) patients, to identify hypoxia in prostate cancer. Methods: PC3 and DU-145 adenocarcinoma cells, in vitro, were gassed with various oxygen concentrations (0–21%) for 24 h, followed by real-time PCR. Xenografts were established in vivo, and the mice were injected with the hypoxic markers [18F]-FAZA and pimonidazole. Subsequently, tumors were excised, frozen, cryo-sectioned, and analyzed using autoradiography ([18F]-FAZA) and immunohistochemistry (pimonidazole); the autoradiograms used as templates for laser capture microdissection of hypoxic and non-hypoxic areas, which were lysed, and real-time PCR was performed. Results: In vitro, all 15 genes were increasingly up-regulated as oxygen concentrations decreased. With the xenografts, all 15 genes were up-regulated in the hypoxic compared to non-hypoxic areas for both cell lines, although this effect was greater in the DU-145. Conclusions: We have developed a combined autoradiographic/laser-guided microdissection method with broad applicability. Using this approach on fresh frozen tumor material, thereby minimizing the degree of RNA degradation, we showed that the 15-gene hypoxia gene classifier developed in HNSCC may be applicable for adenocarcinomas such as prostate cancer.

## 1. Introduction

Hypoxia is a characteristic feature of solid animal [1] and human [2] that occurs because of the structural and functional abnormalities in tumor vessels [2,3] resulting in an insufficient blood supply of oxygen to the growing tumor mass [4]. Generally, tumor hypoxia generates a more aggressive cancer phenotype [5] and causes resistance to both radiation and chemotherapy [6,7]. Therefore, hypoxia is a marker for poor patient/treatment outcome in various cancers. Consequently, several therapeutic modalities to overcome tumor hypoxia have been investigated at both the pre-clinical [7,8] and clinical level [6]. They include treatments to improve oxygen delivery, radiosensitizing hypoxic cells, hypoxic cell cytotoxins, and vascular targeting agents [6,7,8]. It would be beneficial to identify which tumors are “more” or “less” hypoxic to intensify the treatment and justify potential additional side effects. Various methods to determine hypoxia have been evaluated [9]. Exogenous nitroimidazole-based markers such as pimonidazole, identified by immunohistochemistry, and [18F]-labeled fluoroazomycin arabinose (FAZA) for positron emission tomography (PET) scans are promising strategies for the identification of tumor hypoxia [9]. Quantification of endogenous markers expressed by hypoxic tumor cells is another approach for determining the hypoxic status of tumors [10,11]. These have been applied as individual markers [9] or feature a combination of endogenous markers as gene signatures [12]. Our own studies in vitro identified 27 genes from four human squamous cell carcinoma cell lines that were pH-independently up- or down-regulated under hypoxic conditions [13]. Using human primary tumor biopsies from head and neck squamous cell carcinomas (HNSCC) patients with known oxygen status, that signature was refined to include a 15-gene “hypoxia gene expression classifier” [14]. This has retrospectively been shown to have prognostic and predictive impact for the radiosensitizer nimorazole in combination with radiotherapy [15]. The 15-gene hypoxia classifier has also been evaluated in a range of cell lines originating from head and neck, prostate, colon, or esophagus [16], and all 15 genes were found to be up-regulated when comparing anoxic (0% oxygen) with normoxic (21% oxygen) cells in all the cancer cell lines.

Prostate tumors are known to be hypoxic [17]. Studies in which direct pO_2_ measurements in human prostate tumors were performed reported that the more hypoxic tumors were associated with biochemical relapse and local recurrence after treatment [18,19]. Therefore, several attempts have been made to create a clinically relevant prostate-specific hypoxia gene signature. Ragnum and colleagues found a correlation between tumor aggressiveness and pimonidazole immune score [20]. On the basis of these findings, they constructed a 32-gene gene signature that was associated with pimonidazole staining. A validation study in two prostate cancer cohorts showed significant correlations between the gene signature and both Gleason score and survival. Yang et al. created a 28-gene hypoxia gene signature and found that 848 genes were either up- or down-regulated under hypoxic conditions [21]. In a cross-validation study, 28 genes were finally selected, and a further validation from 11 cohorts revealed a significant increase in biochemical recurrence-free survival in radical prostatectomy patients with “low hypoxia” compared to “high hypoxia”. Furthermore, the gene signature was found to have prognostic and predictive impact on radiotherapy treatment and metastatic outcome, respectively.

Marotta et al. described a successful way of profiling hypoxic gene-markers in glioma xenografts using the exogenous hypoxia marker EF5 and a laser-capture microdissection (LCM) apparatus [22]. The EF5 staining detected and visualized the hypoxic tumor areas, which were then dissected, along with the nonhypoxic areas, using the LCM. In our current study, we will demonstrate a refined method with broad applicability where the use of a radio-labeled tracer (in this case [18F]-FAZA) allows autoradiography-guided laser capture microdissection of unstained and unfixed cryo-preserved tumor tissue in vivo (see Figure 1). Furthermore, we investigated the usability of the HNSCC 15-gene signature to monitor hypoxia in PC3 and DU145 prostate cancer cell lines in vitro, and using our refined in vivo method, we investigated the potential of using this hypoxia-induced gene expression in both tumor types grown as xenografts in nude mice.

## 2. Materials and Methods

### 2.1. Cell Lines, Hypoxic Treatment, and Cell Lysis

The prostate adenocarcinoma cell lines (DU-145 and PC-3) were obtained from Dr. Bouchelouche (Department of Clinical Biochemistry, Koege Hospital, Denmark). Cells were cultured in 80 cm^2^ flasks (NUNC) in Dulbecco’s modified eagle medium (DMEM) with GlutaMAX I containing 4.5 g/L D-Glucose, 10% fetal-calf serum, 1% sodium pyruvate, 1% non-essential amino acids, 2% HEPES, and 1% penicillin–streptomycin, with 5% CO_2_ at 90% relative humidity. For hypoxia experiments, 200,000 cells were seeded into 60 mm glass Petri dishes three days prior to experiments, by which time cells were in the log phase of growth. Hypoxia was achieved by continually gassing the cells in an airtight chamber with different oxygen concentrations (0, 0.5, 1, 2, 5, and 21%), 5% CO_2_, and balancing with N_2_, at 37 °C for 24 h. To ensure the appropriateness of our gassing procedure, aerobic indicator strips were included in the chambers when doing oxygen-free incubations. These indicator strips showed that pO_2_ was consistently below 0.15 mmHg for the duration of the experiment. Immediately after removal from the airtight chamber, media was removed, cells were washed with Dulbecco’s phosphate-buffered saline (DPBS), and cells were lysed with Qiazol Lysis Reagent (Qiagen, Hilden, Germany). Cell lysates were stored at −80 °C. Three independent experiments were performed, with duplicate measurements in each experiment.

### 2.2. Animals, Tumor Model, Histology, and Laser Capture Microdissection

The human tumor xenografts, PC3 and DU-145, were grown subcutaneously in both flanks of immune-compromised male NMRI Foxn1nu/Foxn1nu mice. Mice received whole-body irradiation with 4 Gy 1–3 days before tumor cell inoculation to further suppress immunity. For the DU-145 tumor, the number of cells injected was 3 × 10^6^ cells, while for the PC3 cancer, the cell number injected was 2 × 10^6^ cells. Mice were used for experiments when tumors reached a size of 400 to 600 mm^3^, which was typically 6–8 weeks after inoculation. Non-anaesthetized mice were administered intraperitoneally with a mixture of [18F]-FAZA ≈48 MBq (production method described in [23] with minor modifications) in 0.2 to 0.4 mL saline and 60 mg/kg (0.02 mL/g) pimonidazole (for autoradiography and immunohistochemical detection of hypoxia) and returned to their cages. Three to four hours later, mice were sacrificed by cervical dislocation; then, tumors and reference tissue (thigh muscle) were excised and frozen immediately in pre-cooled isopentane (−40 °C, obtained with dry ice). Multiple tissue cryosections were prepared by cutting each tumor xenograft in several layers with 1 mm between each layer. In each layer, consecutive 10 μm sections were cut and either stained for hematoxylin eosin (H&E) or pimonidazole, or used for FAZA autoradiography; the autoradiogram was used as a template for laser capture microdissection (LCM). The sections for LCM were mounted onto Arcturus PEN membrane glass slides (Life Technologies, Naerum, Denmark), and the other sections were mounted on Superfrost^®^ White slides (Menzel-Gläser, Braunschweig, Germany). Immediately after cryosectioning, all tissue sections were processed for autoradiography by exposure to phosphor imaging plates overnight. To minimize mRNA degradation, exposure was conducted at −20 °C. Finally, the intra-tissue tracer signal distribution was extracted at a pixel size of 25 μm using a Fuji BAS 5000 scanner. Tissue sections previously analyzed using autoradiography were either immunologically stained for the distribution of pimonidazole, used to validate the reliability of [18F]-FAZA autoradiograms to visualize hypoxic areas, or H&E stained to evaluate necrosis. By demarcating areas with high [18F]-FAZA accumulation (hypoxic areas) and low [18F]-FAZA accumulation (non-hypoxic areas, defined as areas with [18F]-FAZA activity equal to muscle tissue) from the autoradiography, a printed 1:1 template was made to describe the hypoxic status within the tumor sections. Guided by H&E staining, necrotic areas were avoided, and the demarcated areas were dissected using LCM performed using the Veritas Microdissection Instrument (Arcturus Bioscience) and Capsure Macro LCM caps (Life Technologies). Tissues from LCM representing hypoxic and non-hypoxic areas were lysed immediately after dissection using RLT buffer containing 10 mL/L β-mercaptoethanol.

### 2.3. RNA Extraction, Reverse Transcription, and Gene Expression Quantification

Total RNA was extracted from cell lysates or lysates from laser capture micro-dissected xenograft material using the miRNeasy Mini Kit (Qiagen) according to the manufacturer instructions. A DNase step was included, according to the manufacturer instruction. RNA was eluted in RNAse-free water, and concentrations were determined with QubitTM 3.0 Fluorometer using Qubit^®^ RNA Broad-Range Assay Kit (ThermoFisher Scientific, Waltham, MA, USA). Gene expression levels were quantified using Quantitative Real-Time PCR as described in [13]. Briefly, cDNA was generated using the High-Capacity cDNA Reverse Transcription Kit (Applied Biosystems, ABI, Foster City, CA, USA) according to the manufacturer instructions. Total RNA was reverse transcribed using random primers. Target cDNA transcripts were detected and quantified with TaqMan Gene Expression assays (ABI). For each reaction, TaqMan GeneExpression Master Mix (ABI), cDNA in TE-buffer (Ambion, Austin, TX, USA) and the TaqMan Gene Expression assays were mixed. Quantitative Real-Time PCR was performed on a 7900HT Fast Real-Time PCR System (ABI). Reference genes were ACTR, NDFIP1, and RPL37A; these were previously established as robust reference genes for the 15 gene hypoxia profile [16].

### 2.4. Eppendorf Oxygen-Electrode Measurements

In a separate group of male NMRI Foxn1nu/Foxn1nu mice with the human PC3 or DU-145 xenograft inoculated in the flank, tumor pO_2_ was measured using a 0.35 mm diameter computerized polarographic, oxygen-sensitive needle electrode (The KIMOC-6650 pO_2_ Histograph, Eppendorf, Germany). The electrode was inserted into the tumor and moved automatically through the tissue in 0.7 mm increments, which was followed each time by a 0.3 mm backward step prior to measurement. Four to seven electrode tracks were obtained per tumor, resulting in 52–105 measurements per tumor; the size and shape of the tumor determined the length and number of tracks. Further details of the procedure are described elsewhere [24,25]. The pO_2_ measurements were corrected for tumor temperature, which was determined by inserting a thermocouple needle probe (Ellab Instruments, Copenhagen, Denmark) into the tumor after the pO_2_ measurement. Measurements with negative values were included provided that these were between 0 and −2.0 mmHg, since measurements have an uncertainty of +/− 2.0 mmHg. Values below −2.0 mmHg were discarded as invalid. Data are shown as a histogram from which we selected the median pO_2_ and fraction of pO_2_ values ≤5.0 mmHg; the former value likely reflecting the overall oxygenation status and the latter value radiobiological hypoxia; both parameters have been shown to correlate with radiobiological hypoxia [26].

### 2.5. Immunohistochemistry

The thawed cryosections of interest (two of the 10 sections from each layer) were fixed in a neutral buffered formalin solution (NBF), and endogenous peroxidases were blocked with H_2_O_2_. Then, the sections were rinsed in PBS and slides transferred to LabVision Autostainer 480 (LabVision, Femint, CA, USA). To prevent non-specific background staining, sections were treated with a protein blocking solution (DAKO X0909). Afterwards, the slides were incubated for 40 min with the primary antibody, rabbit polyclonal antibody against pimonidazole (Pab2627 from Hypoxyprobe, Burlington, MA, USA). The primary antibody was detected using the anti-rabbit IgG–horseradish peroxidase-conjugated polymers with a 30 min incubation time (EnVision rabbit reagent, K4003; DakoCytomation, Glostrup, Denmark). Lastly, sections were counterstained with Mayers hematoxylin.

### 2.6. Comparison of [18F]-FAZA Autoradiography and Pimonidazole Images

Pimonidazole and [18F]-FAZA are closely related tracers with a similar binding mechanism. However, since unbound (hypoxia unrelated) pimonidazole is removed during the staining protocol, and binding can be resolved at a cellular level, pimonidazole provides highly reliable information on the distribution of chronically hypoxic cells (“ground truth”). Since autoradiograms contain an unknown amount of unbound [18F]-FAZA, we validated the hypoxia-specificity of [18F]-FAZA by a direct comparison of total tracer signal and density of hypoxic cells. In short, pimonidazole-stained tumor sections were digitalized at a high resolution by the Hamamatsu NanoZoomer 2.0 HT slide scanner (Hamamatsu Photonics, Hamamatsu City, Japan) images were thresholded in ImageJ (NIH, National Institutes of Health, Bethesda, MD, USA), resulting in maps displaying the distribution of viable hypoxic (pO_2_ < 10 mmHg) cells (for further details, see reference [27]). [18F]-FAZA autoradiograms and pimonidazole hypoxia maps were manually co-registered using different landmarks such as tissue section periphery or tissue holes/folds. Following co-registration, the spatial relationship between [18F]-FAZA signal and hypoxic fraction were analyzed by covering the sections with a grid consisting of 1 mm^2^ squares. Only squares fully within the tumor periphery and largely free of necrosis were included.

### 2.7. Data Analysis and Statistics

Correlations between [18F]-FAZA signal intensities and pimonidazole-estimated hypoxic fractions were expressed as Pearson regression coefficients. For the gene expression analysis, thresholds were set manually in the SDS2.1 software. The threshold cycle (CT) values above 35 were regarded as below the detection limit. The gene expression levels of hypoxia-induced genes were calculated using the comparative CT method [28]. ΔCT values were generated by normalizing to the geometric mean of the reference genes ACTR3, NDFIP1, and RPL37A. The fold up-regulation of hypoxic areas compared to normoxic areas was calculated by 2^-ΔCT Hypoxia^/2^-ΔCT Normoxia^. Results from the in vitro gene expression represent data from duplicates of three independent experiments. The results from in vivo gene expression were compiled of 43 samples from non-hypoxic areas and 33 samples from hypoxic areas for the PC3 xenografts, while the results from the in vivo DU-145 xenografts were compiled of 34 samples from non-hypoxic areas and 43 samples from hypoxic areas. Eppendorf oxygen electrode measurements were conducted in single flank tumors in both the PC3 and the DU-145 xenografts. All values are listed as Mean ± 1 Standard Error. Gene expression in hypoxic and non-hypoxic areas of both the DU-145 and the PC3 tumor xenografts were also compared with a Student’s T-test with a *p*-value < 0.05 being considered significant.

## 3. Results

Representative examples of the tumor section images obtained following [18F]-FAZA autoradiography and pimonidazole immunohistochemistry are shown in Figure 2. Comparison of the [18F]-FAZA signal and density of pimonidazole positive cells in the same grid areas increased in a similar fashion, resulting in a linear regression coefficient of 0.8441. Analysis of other tumor sections showed similar positive results with the regression coefficient always around 0.9, thus confirming the potential of using this refined autoradiography and laser-guided microdissection approach to identify hypoxic tumor areas.

In vitro, a hypoxia-induced gene expression in all 15 genes was observed in both the PC3 and DU145 prostate cancer cell lines (Figure 3A,B). It is worth noticing that the fold up-regulation increases as oxygen tension is lowered, but not with the similar intensity in the two cell lines. In the DU-145 cell line, the increase is steeper for the genes EGLN3, KCTD11, P4HA1, and PDK1 when the oxygen tension is decreased from 0.5% to 0% compared to the same genes in the PC3 cell line. The NDRG1 gene expression is much higher in the PC3 cells compared to the DU-145.

Eppendorf oxygen electrode measurements were assessed individually and as a compiled dataset. Oxygenation measurements performed in the PC3 tumor ranged from 0 to 70 mmHg with relative frequencies peaking in the very hypoxic spectrum (Figure 4A). Median values and the fraction of pO_2_ values ≤ 5.0 mmHg, were determined for each individual animal. The mean median value for all animals was 1.7 mmHg (± 0.3 mmHg) and the mean fraction of pO_2_ values ≤ 5.0 mmHg was 73.7% (± 3.6%). For the oxygenation measurements in the DU-145 xenografts, pO_2_ ranged from 0 to 60 mmHg (Figure 4B), with a mean median value of 2.1 mmHg (± 0.6 mmHg), and a mean fraction of pO_2_ values ≤ 5.0 mmHg of 73.0% (±5.6%).

The gene expression levels were quantified in the different fractions of tissue from the sectioning procedure, and the level of the hypoxia inducible genes were compared between the hypoxic and the normoxic areas of the tumor sections. This demonstrated that all 15 genes were significantly up-regulated when comparing hypoxic to non-hypoxic regions in the tumor (Figure 5). A comparison of the genes in the two xenografts revealed an uneven up-regulation. In the DU-145 xenograft, the LOX-gene is nearly 9-fold increased, but it only increased by 1.75-fold in the PC3 xenograft. The size of the standard error of the mean is also larger in the DU-145 LOX-gene expression compared to the LOX-gene expression in the PC3 xenograft, reflecting larger variation in the hypoxic level in the DU-145 xenografts.

## 4. Discussion

We have developed a method that applies laser-guided microdissection in unfixed cryo-preserved tissue sections based on autoradiograms displaying hypoxic and non-hypoxic regions and validated it using a hypoxia gene signature. The method is broadly applicable and can be applied for establishing intra-tumoral spatial links between gene expression and tracers that provides information on microenvironmental (i.e., hypoxia in the present study), metabolism (i.e., FDG/14C-2DG), proliferation (i.e., FLT), and other biological features of interest.

Pimonidazole is a well-known and consistent hypoxic tracer that binds at a partial oxygen pressure <10 mmHg [29]. The pimonidazole adducts can subsequently be visualized using immunohistochemical techniques and is often considered a gold standard. A few studies have reported staining in non-hypoxic tissue such as areas of keratinization in head and neck cancers [30]. In addition, inappropriate antibody levels may reduce the dynamic range of the pimonidazole assay dramatically [31]. Regardless, we, and others, have shown pimonidazole staining patterns typical of chronic hypoxia, with distinct accumulation at a distance from vessels, and in peri-necrotic tissue in a large number of preclinical tumor models [32,33]. Pimonidazole has also been used as a hypoxia marker in prostate cancer patients [34,35,36]. A comparison of [18F]-FAZA autoradiography and pimonidazole images has been done before, revealing a linear relationship between the [18F]-FAZA signal and the density of pimonidazole-positive hypoxic cells [37]. In our experiments, we have made several comparisons of [18F]-FAZA autoradiograms and pimonidazole images, similar to the example shown in Figure 2, and all analyses showed a regression coefficient at approximately 0.90. Since pimonidazole is one of the most verified and validated hypoxic markers, the consistent correlation with [18F]-FAZA autoradiograms reveals a strong association and correspondence between the two methods.

By using fresh frozen tumor material, the RNA degradation is minimal when compared to formalin-fixed and paraffin-embedded (FFPE) tumor samples [38]. The fact that FFPE samples often are exposed to watery solution procedures during immunohistochemical staining protocols worsens the RNA degradation [38]. Marotta et al. stated that their previously developed EF5 fixation method led to RNA degradation [22]. They managed to change the “staining protocol” in order to obtain reliable results, but some degree of RNA degradation was still expected. Ragnum and colleagues described how the index tumor biopsies were FFPE before immunohistochemistry was performed [20]. The validation cohort was from two prostate cancer gene expression datasets, and the tissue fixation protocol of these tumor samples was not described. In the study by Yang et al., the biopsies from seven of the 11 validation cohorts were from FFPE samples, the remainder being fresh frozen tumor samples [21]. Our study, using [18F]-FAZA-autoradiography, enabled us to keep the tumors in an unfixed cryopreserved condition during the whole experiment; from tumor excision until the final steps of dissection and lysing of the hypoxic and non-hypoxic areas of interest.

One study attempted to compare the location of [18F]-FAZA-PET with immunohistochemical staining of hypoxia markers hypoxia inducible factor-1α (HIF-1α) and carbon anhydrase XI (CAXI) to validate [18F]-FAZA-PET as a useful hypoxia identification tool in prostate cancer patients [39]. In none of the 14 patients involved in the study were the tumor nodules visualized by [18F]-FAZA-PET. Nor were the nodules positive for CAIX staining, while the HIF-1α staining was inconsistent. The authors suggested that there might not be any correlation between hypoxia and HIF-1α in prostate cancer, but they could not explain the absence of [18F]-FAZA-PET visualization. Supiot and colleagues used another nitroimidazole-derived PET hypoxia-tracer, namely [18F]-FMISO [40]. In their pilot study, they found that only about 75% of patients with prostate cancer, at any stage, would have [18F]-FMISO-detectable tumors. The small sample size in the Garcia-Parra study [39] could have played a role in their failure to detect [18F]-FAZA-positive tumors. Another factor that could help explain this lack of detection has been suggested [40] and one that involves an underestimation of [18F]-MISO positive areas because of small volumes of hypoxic areas within the prostate tumors; with current PET-scanning techniques, one cannot map volumes less than a diameter of 3 mm. The use of autoradiography in our study allowed us to detect even small [18F]-FAZA-positive areas within the tumors. The inconsistencies in detecting hypoxia in prostate cancer using PET-based approaches does not mean that hypoxia does not exist at meaningful levels in this tumor type or that it is not relevant. Clinical studies in prostate cancer patients using oxygen electrodes [18,19] and the more relevant hypoxic tracer pimonidazole [34,35,36] clearly demonstrated significant levels of hypoxia. Furthermore, these measurements correlated with response to radiation treatment [18,19] and with tumor aggressiveness [34].

A comprehensive analysis of eight published hypoxia signatures in 8006 tumors across 19 cancer types demonstrated a significant correlation between the hypoxia scores generated using the different signatures. Furthermore, the gene expression level of the hypoxia-induced genes were highest in squamous cell carcinomas of the head and neck, cervix and lung, whereas adenocarcinomas of the prostate and thyroid had the lowest level of hypoxic inducible gene expression [41]. This can be interpreted as squamous cell carcinomas of the head and neck being more hypoxic than prostate cancer. However, this must be considered with the reservation that the expression level of some of the hypoxia-inducible genes may be differentially regulated in the different cancer types, even though the genes in the 15-gene hypoxia classifier in vitro have demonstrated to be similarly expressed across different cancer types [16].

Our current study found that all the 15 genes in the hypoxia gene signature were up-regulated in vitro under hypoxic conditions and that the degree of expression increased as the oxygen concentration decreased (Figure 3A,B). This validated the use of the genes for testing the refined microdissection procedure, and applying this approach to the two prostate tumor types, we found similar increased expression of the hypoxia inducible genes (Figure 5A,B), although here, we do not know the exact oxygen concentration. The Eppendorf pO_2_ measurements in the PC3 and the DU-145 xenografts demonstrated that these tumor types are generally hypoxic with the percentage of pO_2_ values ≤ 5.0 mmHg being 74% and 73%, respectively. Oxygen electrode measurements have clearly shown that low tumor pO_2_ values correlate with poor clinical outcome for a review, see [42]. In fact, this has even been seen in prostate cancer [18,19]. The use of oxygen electrodes is generally considered to be a reliable tool to determine hypoxia in tumors, but the method cannot distinguish between acute and chronic hypoxia or low pO_2_ readings due to tumor necrosis. In the study by Milosevic and colleagues [18], direct pO_2_ measurements were made in 247 prostate cancer patients. They found pO_2_ values ≤ 5.0 mmHg in 35% of the measurements and pO_2_ values ≤ 10.0 mmHg in 63% of the readings. The xenografts in our study were grown ectopically (in the flank) compared to an orthotopic location in the prostate gland as in patients in the Milosevic study. Tumor micro-environmental differences due to the location and geno-/pheno-topical dissimilarities may explain the variation in the oxygen partial pressures measured between the clinical study and our pre-clinical evaluation. One study has suggested that pO_2_ values obtained with the Eppendorf histograph were substantially lower in tumors grown in mice compared to human tumors [43], so our findings are not unexpected. What is important is that all 15 genes up-regulated in the in vitro study were also up-regulated in the hypoxic tumor areas, compared to the normoxic areas, in both tumor types. This 15-gene hypoxia classifier was developed in squamous cell carcinoma cell lines and refined in head and neck squamous cell carcinoma patients [13,14]. In the review by Harris and colleagues [12] a total of 32 different hypoxia gene classifiers, in various cancer types, were found following a comprehensive literature search in the most common databases. Nine of the 15 genes in the Toustrup-15-gene hypoxia classifier appeared in the top 20 most frequently appearing genes from all 32 reviewed classifiers. The Toustrup hypoxic gene classifier is also the only classifier that has been directly linked to direct pO_2_ measurements [14]. One study, in which a 28-gene hypoxia gene classifier was created, tried to further validate and compare their findings by testing the Toustrup gene signature on their prostate cancer patient cohorts and found it to be prognostic [21]. Furthermore, the results revealed a strong correlation between their 28-gene scores and the Toustrup signature scores. These findings show that the Toustrup gene-signature is applicable in many tumor types, including prostate cancer.

## 5. Conclusions

The results from this pre-clinical study in PC3 and DU-145 prostate xenografts validate our use of this refined technique for micro-dissecting hypoxic areas in experimental tumors. In addition, the findings confirm the usability of the 15-gene hypoxia classifier in human-derived prostate cancer. Hypoxia is clearly a relevant negative factor in prostate cancer, and further retrospective validation is needed in a clinical setting, with human prostate adenocarcinomas, to fully uncover the potential benefit of using our approach to identify this hypoxia.

## Figures and Tables

**Figure 1 cancers-13-02602-f001:**
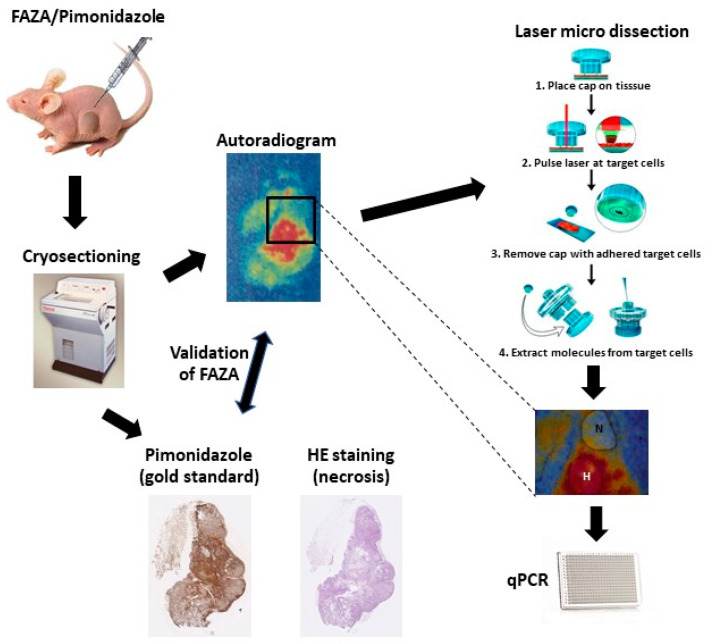
Illustration of the set-up for the in vivo experiment. A nude mouse with a xenograft prostate tumor growing on the flank was intraperitoneally injected with both FAZA and pimonidazole. Approximately four hours later, the mouse was sacrificed, and the tumor was excised and fresh frozen. The tumor was immediately cryo-sectioned in three layers, each layer consisting of 10 consecutive 10 μm sections. Then, these sections were stained for either hematoxylin and eosin (HE) or pimonidazole binding, or used for FAZA autoradiography. The autoradiogram was used as a template for laser micro dissection, where the hypoxic (H) and the non-hypoxic (NH) areas were identified and cut. HE staining was used to evaluate and avoid necrotic areas. This dissected tissue was immediately lysed and real-time PCR (qPCR) performed. The images of the laser micro dissection technique are from Arcturus Bioscience (Mountain View, CA, USA).

**Figure 2 cancers-13-02602-f002:**
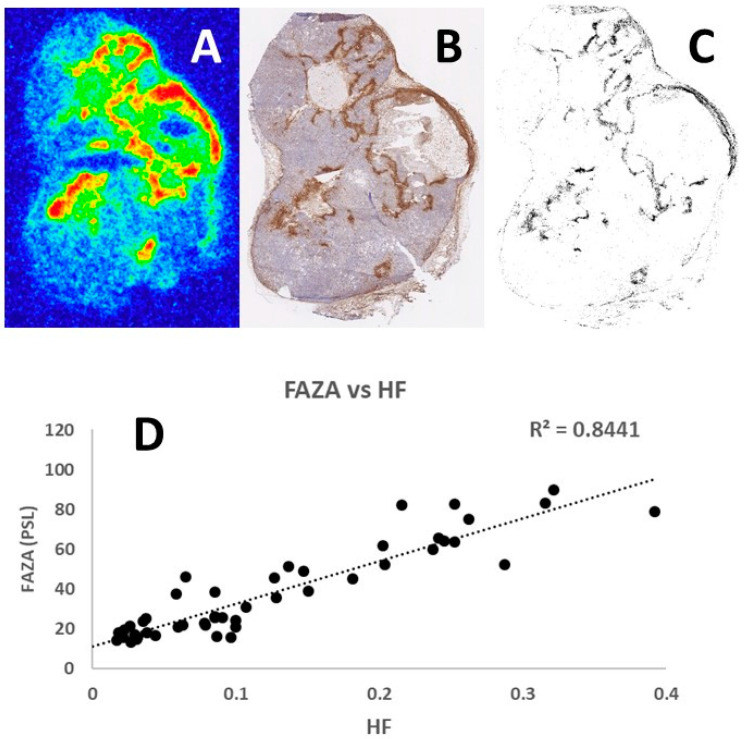
[18F]-FAZA autoradiogram (**A**) and corresponding raw (**B**) and segmented (**C**) pimonidazole staining of the same section (each image is approximately three times larger than the original tumor section), showing that the two hypoxia markers distribute similarly. This was further verified by assessing the spatial relationship between FAZA signal intensity (PSL) and hypoxic fraction (HF), which is derived by placing a grid consisting of 1 mm^2^ squares covering the autoradiogram and the segmented pimonidazole image. Only squares within the tumor periphery and largely free of necrosis were used in the final analysis. (**D**) The resulting correlation between FAZA signal and HF.

**Figure 3 cancers-13-02602-f003:**
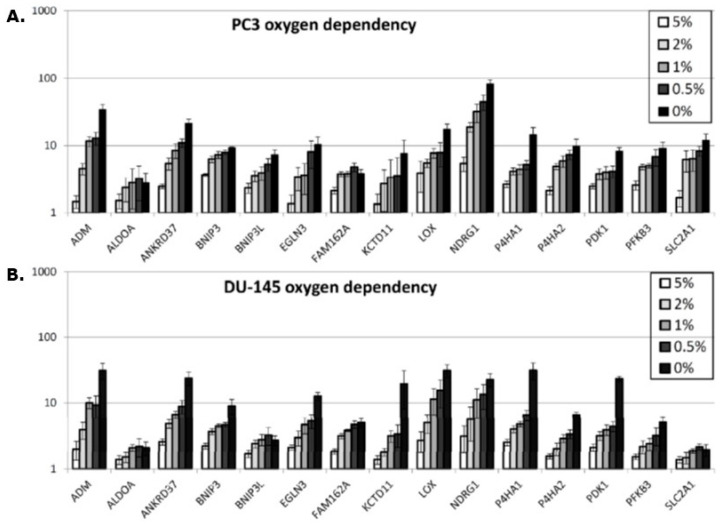
The increased degree of gene expression in vitro under different oxygen concentrations compared to that found under 21% oxygen. Results show means (± 1 S.E.) for *n* = 3 measurements in PC3 (**A**) or DU-145 (**B**) cells. Cells were exposed to the different oxygen concentrations as indicated for 24 h. A greater degree of up-regulation with decreasing oxygen concentration is seen for almost every gene in both cell lines.

**Figure 4 cancers-13-02602-f004:**
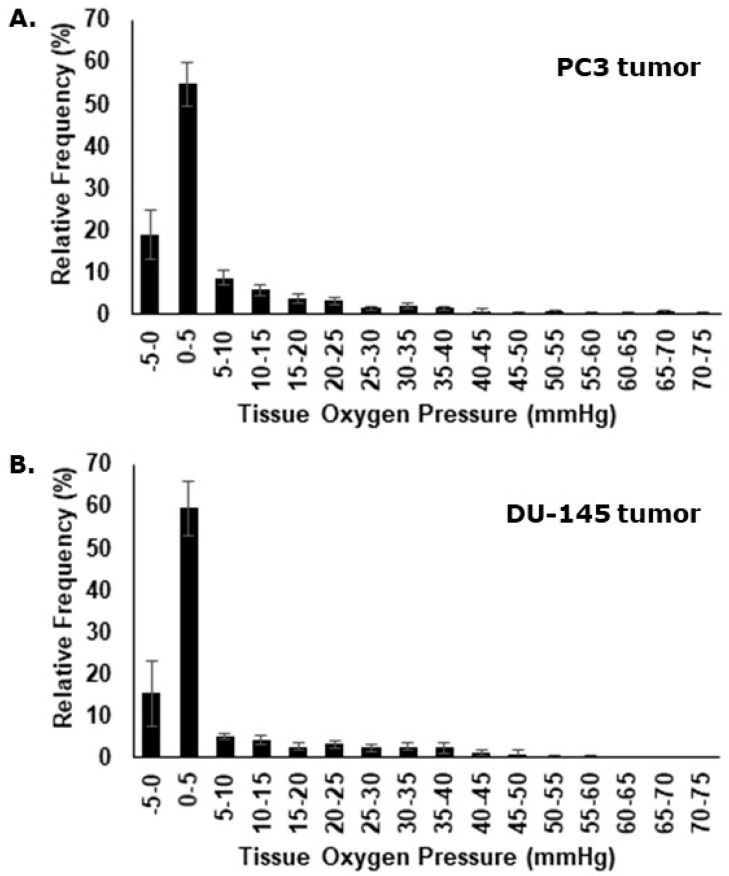
Representative histograms showing the oxygen partial pressure (pO_2_) profiles obtained with the Eppendorf oxygen electrode in PC3 (**A**) or DU-145 (**B**) xenografts. Results show the means (±1 S.E.) of the relative frequency of the various pO_2_ values measured in individual flank tumors from eight (PC3) or 10 (DU-145) tumor-bearing mice. Both tumor types were very hypoxic.

**Figure 5 cancers-13-02602-f005:**
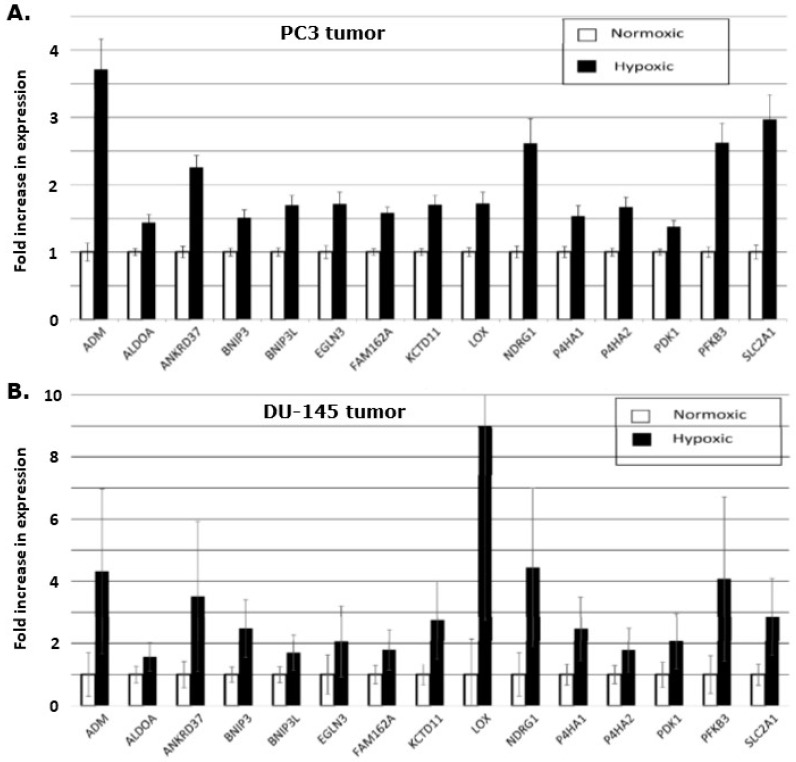
Relative in vivo expression of the 15 hypoxia genes in hypoxic areas when compared to measurements made in non-hypoxic areas. Results are for PC3 (**A**) or DU-145 (**B**) flank xenografts. For the PC3 tumors, the results are based on 43 samples from non-hypoxic areas and 33 samples from hypoxic areas distributed over 17 tumors in nine animals (one animal only had one flank tumor). The results for the DU-145 xenografts consisted of 34 samples from non-hypoxic areas and 43 samples from hypoxic areas (from a total of 12 flank tumors in eight mice). All values are listed as means ± 1 S.E.

## Data Availability

The data presented in this study are available on request from the corresponding author. This data was the product of studies by several different research groups and our Institute does not yet have a universal data depository.

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
