# Peer review of "Refinement of an Established Procedure and Its Application for Identification of Hypoxia in Prostate Cancer Xenografts"

_cancers, 2021, doi:10.3390/cancers13112602_

Round 1

Reviewer 1 Report

The manuscript “Refinement of an established procedure and its application for identification of hypoxia in prostate cancer xenografts” by Pernille Elming and colleagues submitted to Cancers, is focused on an important problem related to the resistance of tumors to treatment caused by hypoxia. The authors have developed a combined autoradiographic/laser-guided microdissection method displaying hypoxic and non-hypoxic regions and validated this approach using a hypoxia gene signature.

This is a well-performed and clearly described work. The results cast no doubt and conclusions are experimentally supported.

The manuscript can be accepted for publication as is.

Minor comment – no error bars in Figure 4

Author Response

Reviewer 1

The manuscript “Refinement of an established procedure and its application for identification of hypoxia in prostate cancer xenografts” by Pernille Elming and colleagues submitted to Cancers, is focused on an important problem related to the resistance of tumors to treatment caused by hypoxia. The authors have developed a combined autoradiographic/laser-guided microdissection method displaying hypoxic and non-hypoxic regions and validated this approach using a hypoxia gene signature.

This is a well-performed and clearly described work. The results cast no doubt and conclusions are experimentally supported.

The manuscript can be accepted for publication as is.

Response to general review: We thank the reviewer for the positive and supportive comments.

Minor comment – no error bars in Figure 4

Response: Error bars have now been added to figure 4 and their explanation added to the figure legend.

Reviewer 2 Report

This is a well-considered study that evaluates the relationship between 18F-FAZA binding and the ‘Toustrup 15-gene hypoxia classifier’ as distinct techniques to detect hypoxia, using two  xenograft models of human prostate carcinoma. The methodology includes cross validated by pimionidazole binding via immunohistochemical detection and direct pO2 measurement by Eppendorf electrode. The study describes improved techniques to concurrently evaluate nitroimidazole-based tissue stains without significant degradation of mRNA signal. The use of laser capture microdissection to delineate 18F-FAZA avid vs naïve regions is elegant.

The Toustrup hypoxia gene signature has previously been validated in head and neck cancer and this work provides preclinical evidence that extends this validation to prostate cancer. This study represents an important step in the cross-validation of techniques to detect hypoxia in prostate cancer, particularly in light of the report by Garcia-Parra et al. which suggested the inability of 18F-FAZA PET to non-invasively detect hypoxia in prostate cancer. 

In summary, this body of work makes an informative contribution to the evaluation of techniques to accurately quantify hypoxia in prostate cancer and their interrelationships.

Minor comments:

Ln 72:  ‘anoxic’ is arbitrarily defined as 0% oxygen. This implies an absolute zero value which cannot be formally true (or proven). Typically, one might use terminology such as less than 10 ppm oxygen (< 10 ppm O2) to describe anoxia, when supported by appropriate direct measurements. If an oxygen-free certified gas bottle is used for gassing boxes this will have been tested to specific limits of residual oxygen and documented by the supplier. Here the gas phase concentration can be defined, whereas and the soluble O2 concentration (in media) remains unknown. Comment extends to Ln 254 and Figure 3. For example, “0%” might be more accurately presented as < 0.001% (or other appropriate value).

Figure 1.  The pimonidazole immunohistochemical staining image is referred to in the subtitle as ‘golden standard’. Arguably the more commonly used term is ‘Gold Standard’. Also, consider whether it is possible to improve the overall resolution of Figure 1?

Ln 111-112: Authors are encouraged to include statement with evidence of cell line authentication e.g. STR analysis. See recommendations and citations therein: https://www.atcc.org/en/Services/Testing_Services/Cell_Authentication_Testing_Service/Cell_Line_Authentication_Test_Recommendations.aspx

Ln 140: verb missing; the autoradiogram was used as a template for laser capture microdissection (LCM).

Ln 192-197: no description of a protein blocking step to minimize artifactual staining of pimonidazole adducts. Was this step unnecessary? Worth clarifying since Figure 1 image shows extensive (whole tumor) DAB positivity. Was this solved by digitalizing image with a Hamamatsu NanoZoomer slide scanner? Figure 2B and 2C appear better quality. Perhaps Figure 1 insert be updated/improved?

Arguably, pimonidazole immunohistochemical staining has had mixed reports, perhaps due in part due to non-specific binding concerns. In the discussion (Ln 293-303) pimonidazole is described as “one of the most verified and validated hypoxic markers”, and whilst it is certainly widely used, perhaps a more balanced (nuanced) perspective might be offered to readers. e.g. DOI: 10.1016/j.radonc.2003.09.012; DOI: 10.1667/RR1305.1; https://doi.org/10.1152/ajprenal.00219.2019;

Figure 4A and 4B: For ease of comparison and interpretation, consider utilizing identical x-axis increments.

Discussion Ln 348 onwards: A comprehensive analysis of the whole-genome correlates of hypoxia in prostate cancer was published in Nature Genetics 2019 (https://doi.org/10.1038/s41588-018-0318-2). This citation arguably deserves some comment as it has implications for the functional interpretation of hypoxic gene signatures. 

Author Response

Reviewer 2

This is a well-considered study that evaluates the relationship between 18F-FAZA binding and the ‘Toustrup 15-gene hypoxia classifier’ as distinct techniques to detect hypoxia, using two xenograft models of human prostate carcinoma. The methodology includes cross validated by pimonidazole binding via immunohistochemical detection and direct pO2 measurement by Eppendorf electrode. The study describes improved techniques to concurrently evaluate nitroimidazole-based tissue stains without significant degradation of mRNA signal. The use of laser capture microdissection to delineate 18F-FAZA avid vs naïve regions is elegant.

The Toustrup hypoxia gene signature has previously been validated in head and neck cancer and this work provides preclinical evidence that extends this validation to prostate cancer. This study represents an important step in the cross-validation of techniques to detect hypoxia in prostate cancer, particularly in light of the report by Garcia-Parra et al. which suggested the inability of 18F-FAZA PET to non-invasively detect hypoxia in prostate cancer. 

In summary, this body of work makes an informative contribution to the evaluation of techniques to accurately quantify hypoxia in prostate cancer and their interrelationships.

Response to general review: We thank the reviewer for the positive and supportive comments.

Minor comments:

Ln 72:  ‘anoxic’ is arbitrarily defined as 0% oxygen. This implies an absolute zero value which cannot be formally true (or proven). Typically, one might use terminology such as less than 10 ppm oxygen (< 10 ppm O2) to describe anoxia, when supported by appropriate direct measurements. If an oxygen-free certified gas bottle is used for gassing boxes this will have been tested to specific limits of residual oxygen and documented by the supplier. Here the gas phase concentration can be defined, whereas and the soluble O2 concentration (in media) remains unknown. Comment extends to Ln 254 and Figure 3. For example, “0%” might be more accurately presented as < 0.001% (or other appropriate value).

Response: The reviewer is totally correct. We have now added the following text to give an accurate value for our definition of anoxia (0% oxygen).

To ensure the appropriateness of our gassing procedure, aerobic indicator strips were included in the chambers when doing oxygen-free incubations. These indicator strips showed that pO2 was consistently below 0.15 mmHg for the duration of the experiment.” (see lines 121-124).

Figure 1.  The pimonidazole immunohistochemical staining image is referred to in the subtitle as ‘golden standard’. Arguably the more commonly used term is ‘Gold Standard’. Also, consider whether it is possible to improve the overall resolution of Figure 1?

Response: The reviewer is correct about the term “Gold standard” and we have corrected this error. A new version of Figure 1 has been included in which the resolution is improved.  

Ln 111-112: Authors are encouraged to include statement with evidence of cell line authentication e.g. STR analysis. See recommendations and citations therein: https://www.atcc.org/en/Services/Testing_Services/Cell_Authentication_Testing_Service/Cell_Line_Authentication_Test_Recommendations.aspx

Response: Both cell lines were commercially purchased before being given to us, so we do not have any statement of authentication, but do not believe there is a potential issue here.

Ln 140: verb missing; the autoradiogram was used as a template for laser capture microdissection (LCM).

Response: The verb has been added as recommended. (see line 145).

Ln 192-197: no description of a protein blocking step to minimize artifactual staining of pimonidazole adducts. Was this step unnecessary? Worth clarifying since Figure 1 image shows extensive (whole tumor) DAB positivity. Was this solved by digitalizing image with a Hamamatsu NanoZoomer slide scanner? Figure 2B and 2C appear better quality. Perhaps Figure 1 insert be updated/improved?

Response: A protein blocking step was used, but we forgot to describe it. The text has now been changed to address this issue as follows:

“To prevent non-specific background staining, sections were treated with a protein blocking solution (DAKO X0909).” (see lines 200-202).

Also, a new version of Figure 1 has been included which has improved the resolution.

Arguably, pimonidazole immunohistochemical staining has had mixed reports, perhaps due in part due to non-specific binding concerns. In the discussion (Ln 293-303) pimonidazole is described as “one of the most verified and validated hypoxic markers”, and whilst it is certainly widely used, perhaps a more balanced (nuanced) perspective mig ht be offered to readers. e.g. DOI: 10.1016/j.radonc.2003.09.012; DOI: 10.1667/RR1305.1; https://doi.org/10.1152/ajprenal.00219.2019;

Response: Again, the reviewer is correct that we should have given a more balanced perspective. To address this issue, we have added the text shown below.

The pimonidazole-adducts can subsequently be visualized using immunohistochemical techniques and is often considered a gold standard. A few studies have reported staining in non-hypoxic tissue such as areas of keratinization in head and neck cancers [30]. In addition, inappropriate antibody levels may reduce the dynamic range of the pimonidazole assay dramatically [31]. Regardless, we, and others, have shown pimonidazole staining patterns typical of chronic hypoxia, with distinct accumulation at a distance from vessels, and in peri-necrotic tissue in a large number of preclinical tumor models [32,33]. Pimonidazole has also been used as a hypoxia marker in prostate cancer patients [34-36].” (see lines 303-311)

However, the reviewer did suggest three additional references, but we have only included two (Janssen et al., 2004 and Koch, 2008) because one dealt with the use of a mouse antibody in rats (Ow et al., 2019) and we don’t believe this was really relevant. Furthermore, to maintain an even balance we have also added two additional references from our own studies that support the use of pimonidazole (Busk et al., 2017 and 2019 – references 32 and 33).

Figure 4A and 4B: For ease of comparison and interpretation, consider utilizing identical x-axis increments.

Response: Figures 4A and 4 B have been modified so the x-axis show identical increments.

Discussion Ln 348 onwards: A comprehensive analysis of the whole-genome correlates of hypoxia in prostate cancer was published in Nature Genetics 2019 (https://doi.org/10.1038/s41588-018-0318-2). This citation arguably deserves some comment as it has implications for the functional interpretation of hypoxic gene signatures. 

Response: This is a very good point by the reviewer and to address it we have now added the following text:

A comprehensive analysis of eight published hypoxia signatures in 8006 tumors across 19 cancer types demonstrated a significant correlation between the hypoxia scores generated using the different signatures. Furthermore, the gene expression level of the hypoxia induced genes were highest in squamous cell carcinomas of the head and neck, cervix and lung, whereas adenocarcinomas of the prostate and thyroid had the lowest level of hypoxic inducible gene expression [41]. This can be interpretated as squamous cell carcinomas of the head and neck being more hypoxic than prostate cancer. However, this must be considered with the reservation that the expression level of some of the hypoxia inducible genes may be differentially regulated in the different cancer types, even though the genes in the 15gene hypoxia classifier in vitro have demonstrated to be similarly expressed across different cancer types 16].” (see lines 365-375)